Sensitivity of soil hydrogen uptake to natural and managed moisture dynamics in a semiarid urban ecosystem

Buzzard Vanessa 1 vbuzzard@arizona.edu
Thorne Dana 1
Gil-Loaiza Juliana 1
http://orcid.org/0000-0003-2952-5129 Cueva Alejandro 2
http://orcid.org/0000-0003-4244-4366 Meredith Laura K. 1 3
1 School of Natural Resources and the Environment, University of Arizona , Tucson, Arizona , United States
2 Biosphere2, University of Arizona , Oracle, Arizona , United States
3 BIO5 Institute, University of Arizona , Tucson, Arizona , United States
Wang Jingzhe
Electronic publication date: 2022 Mar 17
Publication date: 2022
Volume: 10
Electronic Location ID: e12966
Received 2021 Jul 9; Accepted 2022 Jan 28
Copyright: © 2022 Buzzard et al.
Copyright year: 2022
Copyright holder: Buzzard et al.
License: This is an open access article distributed under the terms of the Creative Commons Attribution License, which permits unrestricted use, distribution, reproduction and adaptation in any medium and for any purpose provided that it is properly attributed. For attribution, the original author(s), title, publication source (PeerJ) and either DOI or URL of the article must be cited.
License URL: https://creativecommons.org/licenses/by/4.0/

Keywords: Soil hydrogen uptake, Aridlands, Seasonal precipitation, Water management, Biogeochemistry, Microbial activity, Green infrastructure, Semiarid urban ecosystems, Hydrogen fluxes

Funding: The authors received no funding for this work.

==============================
The North American Monsoon season (June–September) in the Sonoran Desert brings thunderstorms and heavy rainfall. These rains bring cooler temperature and account for roughly half of the annual precipitation making them important for biogeochemical processes. The intensity of the monsoon rains also increase flooding in urban areas and rely on green infrastructure (GI) stormwater management techniques such as water harvesting and urban rain gardens to capture runoff. The combination of increased water availability during the monsoon and water management provide a broad moisture regime for testing responses in microbial metabolism to natural and managed soil moisture pulses in drylands. Soil microbes rely on atmospheric hydrogen (H2) as an important energy source in arid and semiarid landscapes with low soil moisture and carbon availability. Unlike mesic ecosystems, transient water availability in arid and semiarid ecosystems has been identified as a key limiting driver of microbe-mediated H2 uptake. We measured soil H2 uptake in rain gardens exposed to three commonly used water harvesting practices during the monsoon season in Tucson AZ, USA. In situ static chamber measurements were used to calculate H2 uptake in each of the three water harvesting treatments passive (stormwater runoff), active (stored rooftop runoff), and greywater (used laundry water) compared to an unaltered control treatment to assess the effects of water management practices on soil microbial activity. In addition, soils were collected from each treatment and brought to the lab for an incubation experiment manipulating the soil moisture to three levels capturing the range observed from field samples. H2 fluxes from all treatments ranged between −0.72 nmol m−2 s−1 and −3.98 nmol m−2 s−1 over the monsoon season. Soil H2 uptake in the greywater treatment was on average 53% greater than the other treatments during pre-monsoon, suggesting that the increased frequency and availability of water in the greywater treatment resulted in higher H2 uptake during the dry season. H2 uptake was significantly correlated with soil moisture (r = −0.393, p = 0.001, df = 62) and temperature (r = 0.345, p = 0.005, df = 62). Our findings suggest that GI managed residential soils can maintain low levels of H2 uptake during dry periods, unlike unmanaged systems. The more continuous H2 uptake associated with GI may help reduce the impacts of drought on H2 cycling in semiarid urban ecosystems.

Introduction

Atmospheric H2 is an abundant trace gas with a global average of 530 ppb (Schmidt, 1974; Novelli et al., 1999). The primary sources of atmospheric H2 are identified as photochemical oxidation, combustion of fossil fuels, and biomass burning (Novelli et al., 1999). H2 is an indirect greenhouse gas that contributes to climate change by competing for reactions with hydroxyl (OH) radicals in the troposphere (Lee, Rahn & Throop, 2012). If OH radicals react with H2, they are no longer available to react with methane (CH4), a potent greenhouse gas, leading to more atmospheric CH4 (Prather, 2003). H2 is also a source of water vapor production in the stratosphere, which directly impacts ozone (Lee, Rahn & Throop, 2012). It is estimated that half of H2 emissions are from human activity, disproportionately impacting atmospheric H2 concentrations and the H2 cycle of urban ecosystems (Novelli et al., 1999; Ehhalt & Rohrer, 2009). Thus, quantifying soil H2 uptake in urban built landscapes is critical for understanding potential H2 sinks, and the role of natural and water-managed landscapes on H2 cycling in drylands.

Soil uptake is the primary sink for H2 from the atmosphere and is dominated by high affinity hydrogen-oxidizing bacteria (HA-HOB) (Conrad & Seiler, 1979; Conrad, 1996; Novelli et al., 1999; Ehhalt & Rohrer, 2009; Khdhiri et al., 2015; Piché-Choquette & Constant, 2019) that efficiently consume atmospheric H2. Annually, 75% of tropospheric H2 is oxidized by soil microbes for use as an energy source (Schmidt, 1974; Rhee, Brenninkmeijer & Röckmann, 2006). The uptake of atmospheric H2 into soils also contributes to the natural cycling of H2 and has been studied extensively (Smith-Downey, Randerson & Eiler, 2006, 2008; Constant, Poissant & Villemur, 2009; Meredith et al., 2014, 2017; Greening et al., 2015; Khdhiri et al., 2015; Piché-Choquette & Constant, 2019). Both field and laboratory studies have shown that H2 uptake increases with soil temperature and drying (Smith-Downey, Randerson & Eiler, 2006, 2008; Meredith et al., 2017). Although microbial H2 uptake varies during different life stages and is more common during late growth stages and dormancy (Constant et al., 2010; Meredith et al., 2014; Greening et al., 2015), recent studies have shown microbial uptake during active growth (Islam et al., 2020), which may be stimulated by moisture availability in arid and semiarid environments, highlighting the importance of H2 for microbial metabolism in desert biomes (Jordaan et al., 2020).

Arid lands account for roughly 41% of the Earth’s terrestrial surface, support more than one third of the world population, and are projected to expand with changes in climate (Wang, Chen & Dong, 2006; Feng & Fu, 2013; Maestre et al., 2015; Huang et al., 2016; Prăvălie, 2016). In arid and semiarid ecosystems, natural precipitation regimes are important determinants of ecological activity (Noy-Meir, 1973; Huxman et al., 2004). However, changes in natural hydrological processes due to human activity and management may be as important as natural patterns of precipitation on biogeochemical cycling (Austin et al., 2004). As more of the world’s population inhabit and alter drylands, reliance on both natural and managed water sources will also increase.

Tucson, AZ is a city in the Southwest United States which experiences strong seasonal precipitation and increased reliance on green infrastructure (GI) water management practices. The city is home to nearly one million residents and is located in the Sonoran Desert, a semiarid ecosystem with two distinct rainy seasons, one during the winter months and one in the summer (Dimmitt, 2000). Unlike the winter rainy season, the summer rains are caused by the North American Monsoon which is characterized by a seasonal shift in the direction of the prevailing winds from the Pacific to winds from the south and southeast, leading to extreme precipitation events in the Southwest. The intensity of summer rain events accounts for half of the annual precipitation and is met by the reduced ability of urban desert soils to rapidly infiltrate such large volumes of water, which can lead to runoff and flash flooding. To combat stormwater runoff and flooding during the North American Monsoon season, cities like Tucson have implemented GI water management practices to mimic natural hydrological processes by directing and capturing large volumes of rainwater in the soils.

GI techniques include greywater harvesting, rain water harvesting, bioretention basins, and green rooftops. The implementation of these techniques can help mimic the natural water cycle, alter abiotic factors like soil moisture and temperature, and increase microbe-mediated processes that can alter biogeochemical cycling. For example, microbe-mediated fluxes of important trace gases like carbon dioxide (CO2), nitrous oxide (N2O), and methane (CH4) varied with GI design (Grover et al., 2013; McPhillips, Goodale & Walter, 2017; Shrestha, Hurley & Adair, 2018). Moreover, GI designed biofilters commonly used for stormwater management acted as a sink for CH4, and were a small source of N2O during both wet and dry periods with greater CH4 emissions during extreme wet events (Grover et al., 2013). Although soil moisture has been identified as the primary driver of fluxes, soil temperature was also reported as a driver of small and variable soil N2O fluxes and CO2 emissions from roadside bioretention basins (Shrestha, Hurley & Adair, 2018). In addition, flow-through bioretention basins (flat bottom soil bed) using engineered soil media with high phosphorus and C content were sources of CO2, N2O and CH4 (McPhillips, Goodale & Walter, 2017). Although these studies highlight impacts of GI on greenhouse gas emissions, they are constrained to moist temperate climates which may not translate to understanding natural and managed water inputs from water harvested rain gardens in desert urban ecosystems.

GI water management practices combined with the natural precipitation patterns in dryland environments impact soil moisture and temperature which directly impact biogeochemical cycling (Buzzard et al., 2021). Rain gardens irrigated with harvested water are common GI techniques that alter the nutrient inputs, and the frequency and quantity of wetting events. The soil moisture legacy in GI systems is also dependent on the local monsoonal precipitation regime, which brings cooler temperature and ensures water is replenished for irrigation during periods of drought. These changes in precipitation imposed by this natural and managed moisture regime provide a unique opportunity to directly measure these critical environmental drivers to better understand the role of soil H2 uptake in a dryland environment. In this study, we use a residential GI rain garden to assess how seasonal precipitation and irrigation from harvested water sources impact H2 fluxes. To test the combined effects of seasonal precipitation and GI water management on soil H2 fluxes, we ask, (i) how does the precipitation regime, specifically monsoons, of the Sonoran desert, influence H2 fluxes?; and (ii) do different GI water harvesting techniques affect soil H2 fluxes? Greater H2 uptake has been observed in ecosystems with high temperature and variable soil moisture, highlighting the sensitivity of H2 to soil moisture availability in water-limited environments (Conrad & Seiler, 1985). Therefore, we hypothesize that increased water availability observed during the monsoon season leads to greater H2 uptake and that soils from GI water harvesting rain gardens results in more H2 uptake during the dry season when managed basins continue to receive water through regular irrigation.

Materials and Methods

Study system

Our study site is located at a residence in central Tucson, Arizona-with a mean annual temperature of 17 °C and mean annual rainfall is 32.2 cm (Sheppard et al., 1999). In November of 2017, a green infrastructure (GI) water harvesting system was installed by Watershed Management Group Inc. (Tucson, Arizona) at our study site; site pictures and study design are described in (Buzzard et al., 2021). Specifically, three rain garden basins were dug to direct and capture rainwater and a 3,000 gallon plastic rain storage tank was installed to capture rooftop runoff, as described in (Buzzard et al., 2021). Three GI treatment plots (1 m × 4 m) received different water inputs and sources (passive stormwater runoff, active irrigated with tank-stored harvested rainwater, and greywater from laundry) along with an unaltered control treatment area that represented the initial, flat pea gravel-lain condition. All treatments were composed of four 1 m2 batches within a 4 m by 1 m plot that received limited municipal city irrigation to establish plants and were subjected to the same rainfall, but differed in the inputs and frequency of irrigation from harvested water sources, and the type and depth of the mulch layer as described in (Buzzard et al., 2021). Specifically, the greywater treatment was irrigated with residential laundry water effluent and rain collected in the storage tank. The active water treatment received rainwater irrigation from a rain storage tank that collects and stores roof-top runoff, and may also receive overflow from the tank during large rain events that can flood the basin. The passive treatment was a dug basin designed to collect stormwater runoff and overflow from the active basin, yet both the passive and control (flat-elevated landscape) water treatments received primarily incidental rain.

Soil monitoring system

We installed a meteorological station to measure microclimate variation at the site. We collected local air temperature at 3 m above ground using a climate sensor (VP-4, METER Group, Inc., Pullman, WA, USA) and rain gauge to measure precipitation (ECRN-100, METER Group, Inc., Pullman, WA, United States). The onsite rain gauge did not work between the 165th and 239th day of the year, and precipitation data from a nearby weather station were used to estimate daily precipitation in the area (“AZMET: The Arizona Meteorological Network: Tucson Station Data Files”, https://cals.arizona.edu/AZMET/01.htm). In each treatment, we installed soil moisture and temperature sensors (ECH2O 5TE, METER Group, Inc., Pullman, WA, USA) at 12.5 cm belowground in the center of the treatment, connected to data loggers (EM50, METER Group, Inc., Pullman, WA, USA) collecting hourly data from January 2018 to December 2018 (Buzzard et al., 2021).

Soil hydrogen fluxes

Soil H2 fluxes were measured using vented static chambers during four campaigns (1) pre-monsoon (June 8th), (2) mid-monsoon (August 2th), (3) late-monsoon (September 12th), and (4) post-monsoon (November 14th), in response to variation in soil moisture throughout the monsoon season and to assess variation in microbial activity across GI treatments. For example, the pre-monsoon campaign was characterized by high temperature and low soil moisture; the campaigns during mid-monsoon and late-monsoon season were characterized by high temperature and high soil moisture; and the post-monsoon campaign was characterized by lower temperature and intermediate (moist) soil moisture. To estimate H2 fluxes, four cylindrical PVC collars (20 cm diameter, 20 cm height, surface area = 0.0314 m2) were installed 10 cm into the soil within each treatment 1 week prior to sampling. All gas sampling was performed between 8:30 and 10:30 a.m. to reduce the effects of high temperature experienced during the summer months in Tucson, AZ. The mulch layer was removed from the soil collar 2 h prior to sampling to reduce the effect of different mulch types and depths between treatments and allow a direct assessment of the soil surface. Static vented chambers were placed on the soil collar approximately 2 cm to ensure a proper seal and consistent headspace. Immediately following chamber placement, 5 ml gas samples were collected using air-tight syringes from the sampling port at four time points (0, 5, 10, and 15 min). All four treatments were sampled at the same time with randomization of samplers (people) and chambers between each batch and treatment to reduce operator-sampling biases. Syringes were placed in a cooler to maintain temperature and reduce exposure to ultraviolet light until they were transferred to the lab. A total of 64 gas samples were collected during a single morning for each time period, totaling 256 gas samples over the monsoon season. H2 was measured in parts per billion using a reducing compound photometer (RCP model 910-105 series, Peak Laboratories LLC, CA, USA) gas chromatography (GC) within 24 h of sampling. Each gas sample was measured in triplicate from 1 ml gas injections. Ultra zero grade air was used as the carrier gas. Chamber height and temperature were measured during field sampling and used to calculate H2 fluxes for each sampling event based on the following equation:

F=(ΔC/Δt)∗(PaV/RT)∗(1/A)

where F is the H2 flux (nmol H2 m−2 s−1), ΔC/Δt is the rate of change of the H2 concentration in the headspace of the chamber through time, Pa is the atmospheric pressure (atm), V is the total chamber volume (L) measured for each chamber and sampling event, R is the gas constant (0.08206 atm L mol−1 K−1), T is the temperature of air inside the chamber (K), and A is the surface area covered by the chamber (0.0314 m2). It is important to note that negative F represents H2 uptake into the soil, whereas positive F values represent emission of H2 into the atmosphere, so greater rates of H2 uptake correspond to more negative soil H2 flux values.

Soil moisture incubation experiment

To determine if moisture was the dominant influence on H2 fluxes, we conducted a soil microcosm experiment that altered soil moisture levels but maintained constant temperature. Soil samples from each batch (n = 4) within each treatment (n = 4) were collected during the post-monsoon sampling and were air dried for approximately 7 days until they had a base soil moisture level of roughly 2% gravimetric water content (GWC). GWC was calculated for all air dried samples and water was added to increase the GWC to the three target levels; dry corresponded to the low range of observed GWC values (2%), moist is the average range observed (10%), and wet corresponded to the wet end of the range (20%). Each group represented conditions observed in the field during the 2018 monsoon season. Soils were then incubated at 25 °C for 7–8 days and then H2 fluxes were measured for three different moisture levels within 24 h. Moisture levels were maintained during incubation by limiting evaporative loss by covering vials with parafilm wax. Samples were randomized and vials were closed using a rubber stopper for roughly 20 min, where 1 ml of air was removed using an air tight syringe at regular intervals during the 20 min period to calculate fluxes and directly injected into the GC-RCP to measure H2 concentrations. Empty serum vials were used as a control and measured during the experiment.

GC calibration

We used a tank of breathing grade air (Al B300; Airgas, Radnor, PA, USA) as a working standard for this study. We estimated the amount of H2 in the breathing air tank to be 664 ppb H2 by comparison to atmospheric H2 measurements. Specifically, we used the median H2 peak areas in breathing air and atmospheric gas measurements and assumed a representative value for typical atmospheric hydrogen concentration of 530 ppb. For each round of field and lab H2 measurements, we used the median of at least 21 breathing air tank peak areas measured over the course of the sampling event to calculate a GC response factor ([atmospheric H2]/[GC H2 peak area]). The response factor was used as a scaling factor for single-point calibration of all unknown samples for each sampling event. This approach allowed us to account for possible drift in the GC response factor between measurement days, which ranged from a median of 0.00306 ppb area−1 to 0.00361 ppb area−1. The response factor varied by 5.4% (stdev) across all sampling periods and maintained a smaller variation than the uncertainty of the GC-RCP analyzer of 10%.

Statistical Analyses

All statistical analyses and figures were completed in R (R Core Team, 2021). Daily averages for soil moisture and temperature from each treatment were calculated for the four in situ flux sampling events. The gvlma package in R was used to assess if the model met statistical assumptions (Peña & Slate, 2006). Visual inspection of residual plots did not reveal any extreme outliers or obvious deviations from homoscedasticity or normality for soil H2 fluxes. We used a two-way mixed analysis of variance (ANOVA) to assess the effects of treatment and season on soil H2 fluxes. Effect size was determined by generalized eta squared (ges) with small (ges = 0.2), medium (ges = 0.5), and large (ges = 0.8) following Cohen suggested values (Cohen, 1992). Pairwise comparisons were assessed using Tukey’s honest significant difference post hoc methods. Soil H2 fluxes from the microcosm experiment did not meet statistical assumptions for kurtosis. However, results did not differ between transformed and non-transformed H2 fluxes data from the microcosm, thus nonparametric one way analyses were completed on non-transformed data. A p-value of <0.05 with Bonferroni adjustment was used to determine statistical significance. Generalized linear regression was used to assess the individual effect of soil temperature and soil moisture on in situ H2 fluxes.

Results

Meteorological conditions

Meteorological data were used to measure local climate variability over the monsoon season (Fig. 1; Table S1). In 2018, we recorded 342 mm of precipitation and an average air temperature of 22.88 ± 6.34 °C at our site. The average daily air temperature in our study site was similar for the first three sampling dates (i.e., pre-monsoon, mid-monsoon, late-monsoon), ranging from 32.2 to 33.6 °C, but was much lower during the post-monsoon sampling (14 ± 3.62 °C). No precipitation was recorded in the week prior to each sampling event, with the exception of the mid-monsoon sampling, which received 6.86 mm of rain the day before sampling. However, prior to that rain event, conditions were relatively dry with no observed precipitation for the previous 10 days. Soil and air temperature followed a similar temporal pattern with the highest temperature recorded between May and October (Fig. 1A; Tables S1 and S2), and the greatest fluctuation in soil temperature was observed in the control treatment (Buzzard et al., 2021).

Figure 1 Meteorological data measured hourly on site.

(A) Site level air temperature (°C) measured at 3 m aboveground. Light grey represents the hourly temperature. Dark grey line represents the smooth fit function from a generalized additive model estimation of air temperature. (B) Daily cumulative sum of precipitation (mm) at the site. Dashed red lines represent the static chamber hydrogen sampling dates.

In situ soil hydrogen fluxes

We found that the interaction between green infrastructure (GI) treatment and increased wetting events observed during the monsoon season significantly affected soil H2 fluxes (Fig. 2; Table 1; F(9, 48) = 2.72, p = 0.012, ges = 0.338). In addition, the mean H2 fluxes from each treatment differed across the four sampling points, with pre-monsoon and late-monsoon H2 fluxes statistically different between treatments. During the pre-monsoon sampling period we observed significantly higher H2 uptake into the soil in the greywater treatment compared to the passive and control treatments (Table S3). Whereas the passive treatment had a significantly lower uptake into the soil compared to the other treatments during the late-monsoon (Table S3).

Figure 2 The effects of green infrastructure management and time during the monsoon season on soil hydrogen fluxes (nmol H2 m−2 s−1).

Boxplots displayed with point distributions for each batch within a treatment. The center line represents the median, and the lower and upper lines correspond to the first and third quartiles (25% and 75% quartiles). Whiskers correspond to the 95% confidence intervals. The two-way ANOVA presented F statistic, p-value and generalized eta squared (ges). The grey dashed line at 0 on the y-axis represents the transition between soil uptake (negative values) and emission into the atmosphere (positive values).

Table 1 Analysis of variance (ANOVA) assessing the effects of green infrastructure management and soil moisture on hydrogen fluxes (nmol H2 m−2 s−1).

Experiment location	Effect	DFn	DFd	Test statistic	p-value	p < 0.05	Effect size	
Field	Treatment	3	48	4.64	0.006	*	0.225	
Field	Season	3	48	29.85	0.000	*	0.651	
Field	Treatment:Season	9	48	2.72	0.012	*	0.338	
Lab	Treatment	3	48	1.40	0.706		−0.036	
Lab	Moisture level	2	48	34.35	0.000	*	0.719	
Note:

Two-way ANOVA for assessing hydrogen fluxes as a function of treatment and season for in situ field sampling. One-way nonparametric Kruskal-Wallis ANOVA independently assessing hydrogen fluxes as a function of treatment and moisture level for microcosm lab experiment. DFn, Degrees of freedom in the numerator. DFd, Degrees of freedom in the denominator. Test Statistic: F for two-way parametric ANOVA; H for one-way nonparametric Kruskal-Wallis test.

Soil microcosm hydrogen fluxes

We used a microcosm experiment to test the direct effect of different soil moisture levels on soil H2 fluxes and did not find a significant difference between treatments (Fig. 3; H(3, 48) = 1.4, p = 0.706). However, the mean H2 fluxes were statistically different between the three soil moisture levels observed for each treatment (Table 1; H(2, 48) = 34.35, p < 0.001), with H2 uptake significantly greater at wet moisture levels (20% gwc) than dry (2% gwc) and moist (10% gwc) soil moisture levels (Table S4).

Figure 3 Microcosms assessing the effects of green infrastructure management and soil moisture on hydrogen fluxes (nmol H2 m−2 s−1).

Boxplots displayed with point distributions for each batch within a treatment. The center line represents the median, and the lower and upper lines correspond to the first and third quartiles (25% and 75% quartiles). Whiskers correspond to the 95% confidence intervals. The one-way nonparametric Kruskal-Wallis ANOVA was performed independently for both treatment and moisture level on H2. Corresponding p-values are recorded with treatment first and moisture level second. The grey dashed line at 0 on the y-axis represents the transition between soil uptake (negative values) and emission into the atmosphere (positive values).

Abiotic drivers of hydrogen fluxes

Soil temperature and moisture are known drivers of microbial diversity and activity. To test the independent effect of temperature and moisture on H2 fluxes we assessed the relationship between soil H2 fluxes from the in situ measurements independently with soil temperature and moisture. We found an inverse effect of soil temperature and moisture on soil H2 fluxes. Soil H2 uptake decreased with increased temperature (r = 0.345, p = 0.005, df = 62; Fig. 4), and increased with increased soil moisture (r = −0.393, p = 0.001, df = 62; Fig. 4).

Figure 4 Effect of selected environmental drivers during in situ measurements of soil hydrogen fluxes (nmol H2 m−2 s−1).

(A) Positive correlation of hydrogen fluxes and soil temperature °C shows H2 uptake into the soil is greater at lower temperature, and (B) negative correlation between hydrogen fluxes and soil moisture indicates that H2 uptake increases with increased moisture. Boxplots display the median, 25th and 75th interquartile range, and whiskers show 1.5 times the interquartile for soil temperature (top left), soil moisture (top right), and soil hydrogen fluxes (right side) averaged during the in situ sampling period. Negative H2 fluxes indicate uptake from the atmosphere into the soil.

Discussion

The primary goal of this study was to examine the interactive effects of green infrastructure (GI) practices and seasonal precipitation on soil H2 fluxes in a semiarid urban ecosystem. We hypothesized that changes in soil microclimate driven by implementation of GI, specifically changes in soil moisture variability, would lead to increased H2 uptake in the soils that would be further amplified by seasonal variability in precipitation during the monsoon season. The field observations suggested that soil H2 uptake was triggered by increased precipitation during the seasonal monsoon across all treatments. In addition, soil H2 uptake was greater in the greywater treatment compared to the other treatments during pre-monsoon, suggesting that the increased frequency and water availability in the greywater treatment resulted in higher H2 uptake during the dry season and may support greater microbial consumption of H2 consistently throughout the year. H2 uptake was significantly correlated with soil moisture and temperature, which are key environmental factors impacted by both the seasonal precipitation regime and GI water management practices.

The North American Monsoon triggers biological activity by bringing moisture and cooler temperature to hot-dry soils. However, the large quantities of rain with each storm may lead to flooding and saturation of soils, reducing diffusivity of gases and these factors coupled with lower infiltration rates and sealing of urban soils limit or decrease soil H2 uptake. H2 uptake into soils is limited by abiotic and biotic processes under dry and wet soil conditions. For example, under dry conditions H2 diffusion into soils is biotically limited by the presence of dry inactive layers with reduced biological activity in desert soils (Fallon, 1982; Conrad & Seiler, 1985; Smith-Downey, Randerson & Eiler, 2008; Bertagni, Paulot & Porporato, 2021); while wet soil conditions reduce movement into the soils, also leading to H2 diffusion limitation (Yonemura, Kawashima & Tsuruta, 1999; Yonemura & Kawashima, 2000; Gödde, Meuser & Conrad, 2000; Ehhalt & Rohrer, 2013). During the pre-monsoon season, H2 uptake was lower in the control and passive treatments, with greater uptake in the active and greywater GI treatments, consistent with research showing that H2 uptake even at low moisture levels may promote microbial activity (Fallon, 1982; Conrad & Seiler, 1985; Smith-Downey, Randerson & Eiler, 2006, 2008). Additionally, studies have shown that under very low soil moisture conditions, HOB activity is drastically inhibited (Paulot et al., 2021). As the monsoon season progressed, increased soil moisture resulted in greater H2 uptake across all treatments, suggesting semiarid urban ecosystems are sensitive to changes in moisture availability and that there is increased uptake throughout the monsoon season as long as there are continual rain events.

As temperature decreased during the post-monsoon season, we observed median moisture levels and similar soil H2 uptake across treatments, suggesting an ideal moisture range between 15% and 18% near 15 °C, which is within the ideal temperature range observed for microbial H2 uptake in the Mojave Desert (Smith-Downey, Randerson & Eiler, 2006). While there are models that robustly assess temperature and moisture effects on H2 uptake, our results oppose current research that show H2 uptake increases with increased temperature which may correspond with limitations in our dataset (Ehhalt & Rohrer, 2011, 2013; Yashiro et al., 2011; Bertagni, Paulot & Porporato, 2021). Specifically, our data capture a narrow range of temperature (12 to 39 °C) compared to the broad range (−20 to 100 °C) modeled in (Ehhalt & Rohrer, 2011) and are missing a critical range between 16 and 27 °C which has been identified as an optimal temperature range of soils with 19% moisture for peak H2 uptake in the Mojave Desert (Smith-Downey, Randerson & Eiler, 2008). In fact, our results suggest greater variability in H2 uptake at higher temperature, with less uptake measured in the passive and control treatments during the driest sampling period (pre-monsoon). Our findings highlight the interaction of temperature and soil moisture as important abiotic drivers of H2 uptake in semiarid managed systems, with GI altered soil moisture or temperature leading to reduced seasonal variability but sustained H2 uptake.

The widespread implementation of GI water management practices alter soil moisture legacy by irrigating rain garden basins at different frequencies and providing a more constant water source during the dry season. More frequent and consistent soil moisture events in the greywater compared to the other treatments can lead to shifts in the soil microbiome (Buzzard et al., 2021), nutrient availability, and carbon inputs. Desert soils have low carbon availability, and changes in the water inputs from GI may increase soil carbon by supporting vegetative growth, root development, and increased soil microinvertabrate diversity and abundance (Pavao-Zuckerman & Sookhdeo, 2017). The sustained availability of water and organic resources for energy may further support organoheterotrophic soil microbes, such as Actinobacteria, Proteobacteria, and Chloroflexi (Lynch et al., 2014; Leung et al., 2020; Jordaan et al., 2020), leading to metabolically flexible microbiomes in GI treatments capable of more rapidly switching between resources (Bay et al., 2021a, 2021b). In addition, soil H2 uptake models project that greater HA-HOB activity is associated with higher temperature, but can be limited if the minimum soil moisture threshold is not met (Paulot et al., 2021), and further research should address the uncertainty in HA-HOB activity in arid and semiarid urban ecosystems. As observed in the greywater basin, our findings suggest that a more consistent soil moisture legacy was important for microbial-mediated biogeochemical processes and highlights a sensitivity to increased soil moisture in semiarid urban ecosystems.

Atmospheric trace gases, like H2, are important for sustaining soil microbes during periods of low resource availability and may also act to alleviate competition between organoheterotrophs (Fierer, 2017; Bay et al., 2021a, 2021b). Microbial communities with more diverse metabolic strategies may be more resilient to changes in climate and anthropogenic disturbances (Meyer et al., 2004; Allison & Martiny, 2008; Berney & Cook, 2010; Greening et al., 2014). As climate changes, shifts in seasonal precipitation patterns may lead to longer dry periods and more variable precipitation patterns in arid and semiarid regions of the Southwestern United States (Zhang et al., 2021). Although recent work suggests that H2 uptake may increase with warmer temperature observed at midlatitudes in the northern hemisphere, altered soil carbon and reduced soil moisture may reduce microbial activity resulting in unknown feedbacks in atmospheric concentrations (Paulot et al., 2021). Here, we show that water management practices, like GI, buffer temperature extremes and reduce periods of drought, increasing H2 uptake. Our findings suggest that GI managed residential soils can maintain low levels of H2 uptake year around, above what would be sustained in unmanaged systems (i.e., our control treatment), and may reduce the impacts of drought on H2 cycling in semiarid urban ecosystems.

Conclusions

Atmospheric H2 is impacted by changes in climate and increased anthropogenic emissions. Characterizing the changes in microbial mediated soil H2 uptake has important implications for semiarid urban systems where local land management practices alter the abiotic and biotic environment, increasing uncertainty in the atmospheric hydrogen cycle. Here, we showed that environmental factors impacted by seasonal precipitation and GI water management practices in the semiarid deserts of Arizona increased soil H2 uptake during seasonally dry periods. As temperature and drought increase in the Southwestern United States, uptake of atmospheric H2 by soils may be reduced, unless cities continue to implement GI water management practices, which decrease dry periods and support greater H2 uptake. Continued investigation into how managed systems impact soil microbial community composition and function is suggested to better understand the distribution and functional role of HA-HOB microbial communities in sustaining H2 uptake in semiarid urban climates during drought. Our findings highlight the interaction between natural and managed water regimes on soil H2 uptake as corroborated by field and lab measurements, reinforcing the unique role of H2 in microbial metabolism in semiarid urban landscapes.

Supplemental Information

Supplemental Information 1 Daily summary of meteorological data for 10 a day prior to in situ sampling.

Cumulative Daily precipitation in mm, daily average temperature with the standard deviation. Blue, starred text notes sampling day.

Click here for additional data file.

Supplemental Information 2 Mean and standard deviation data for soil moisture and temperature at 12.5 cm depth for a 10 day prior to in situ sampling.

Blue, starred text notes sampling day.

Click here for additional data file.

Supplemental Information 3 Post hoc comparison of hydrogen fluxes between treatments by season.

Pairwise comparisons using Tukey’s HSD (honest significant difference) with Bonferroni correction.

Click here for additional data file.

Supplemental Information 4 Post hoc comparison of hydrogen fluxes at each moisture level.

Pairwise comparisons using Dunn Test with Bonferroni correction.

Click here for additional data file.

Supplemental Information 5 Precipitation data.

Click here for additional data file.

Supplemental Information 6 Air temperature.

Click here for additional data file.

Supplemental Information 7 Microcosm soil H2 flux data.

Click here for additional data file.

Supplemental Information 8 I. situ soil H2 fluxes.

Click here for additional data file.

Supplemental Information 9 R-script for figure 1.

Click here for additional data file.

Supplemental Information 10 R-script for figures 2 and 3.

Click here for additional data file.

Supplemental Information 11 R-script for figure 4.

Click here for additional data file.

We would like to thank Erik Arcos, Andreas Brændholt, and Leslie Dominguez for help with field sampling; and Peter Moma and L. Tate Montgomery for laboratory assistance.

Additional Information and Declarations

Competing Interests

Author Contributions

Data Availability

The authors declare that they have no competing interests.

Vanessa Buzzard conceived and designed the experiments, performed the experiments, analyzed the data, prepared figures and/or tables, authored or reviewed drafts of the paper, and approved the final draft.

Dana Thorne conceived and designed the experiments, performed the experiments, analyzed the data, authored or reviewed drafts of the paper, and approved the final draft.

Juliana Gil-Loaiza conceived and designed the experiments, performed the experiments, authored or reviewed drafts of the paper, and approved the final draft.

Alejandro Cueva conceived and designed the experiments, authored or reviewed drafts of the paper, and approved the final draft.

Laura K. Meredith conceived and designed the experiments, analyzed the data, authored or reviewed drafts of the paper, and approved the final draft.

The following information was supplied regarding data availability:

The data and r-scripts are available in the Supplemental Files and at

OSF: Buzzard, Vanessa. 2022. “Sensitivity of Soil Hydrogen Uptake to Natural and Managed Moisture Dynamics in a Semiarid Urban Ecosystem.” OSF. February 22. DOI 10.17605/OSF.IO/QA7HZ.

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
