# Peer review of "Sensitivity of soil hydrogen uptake to natural and managed moisture dynamics in a semiarid urban ecosystem"

_PeerJ, doi:10.7717/peerj.12966_

## Round 0.1 · original submission · Major Revisions

The reviewers' comments on your work have now been received. The referees acknowledge the potential interest of your work, but between them, they also raise a number of concerns, which must prevent us from offering to publish the paper in its present form. The referees’ reports seem to be quite clear. Naturally, we will need you to address all of the points raised. The writing is not clear enough, and there are some issues.

Reviewer 1 ·

Basic reporting

This article reports on H2 soil uptake measurements in rain gardens exposed to three water harvesting practices in arid rain gardens. The potential ecological legacy of water harvest practices on the relationship between H2 soil uptake activity and soil moisture was also examined in a laboratory incubation experiment. The experimental design elaborated by the authors was straightforward and elegant, providing tangible results regarding environmental drivers shaping the biological oxidation activity of H2 in soil. This will contribute to the body of knowledge we need to steer soil microbial communities towards the provision of ecosystem services in built and natural environments.

The text is clear and focused and results are aligned with posited hypotheses. The quality of figures, tables and supplementary information is impeccable. The article includes sufficient introduction and background information related to the biological sink of atmospheric H2 and green infrastructure to demonstrate the relevance of the investigation. Citations were carefully selected throughout the manuscript, but two exceptions:

i) Contrary to the information conveyed by the authors in the abstract (L36-38) and the introduction (L140-141), simulation of H2 uptake activity in arid ecosystem caused by soil wetting is not novel. This phenomenon was reported in the pioneering work of Conrad & Seiler in 1985 (J Geophys Res). This should be acknowledged in the text.

ii) Discovery of high-affinity H2 oxidation activity in the dormant life stage of actinobacteria was reported by Constant et al., 2010, Environ. Microbiol. That citation must be included in the text.

Experimental design

In general, methods are described with sufficient detail information to replicate. The renowned expertise of the research group in H2 and trace gases flux measurements is supported by the rigor regarding treatment replication, gas sampling and analyses. I however recommend a few clarifications :

(i) Description of the experimental design is vague in a statistical point of view. I understand that four replicated plots were included in the study, but the text is not explicit regarding their spatial organization. Was a complete random design or a randomized complete block design implemented in the field? What is the distance between the plots? A schematic representation of the field trial would be valuable to address these questions.

(ii) More information is necessary to better distinguish control and passive irrigation treatments. Besides woodchips (passive) and pea gravel (control) substrates on the ground, what are the differences between both treatments?

(iii) More details are necessary to describe the calibration procedure of the reducing compounds photometer gas chromatography system (i.e. standard gas).

Validity of the findings

Besides correlations of H2 soil uptake rate with soil moisture and temperature, this article provides evidence supporting benefits of green infrastructures to alleviate disturbance of soil microbial processes in arid ecosystems. Promotion of H2 soil uptake activity triggered by soil irrigation with grey water is supported by robust experiments and statistical analyzes. Raw data has been provided and they are robust.

Additional comments

I have a few minor comments to conclude my report.

(i) Text commenting correlations between soil temperature (and soil moisture) and H2 fluxes is erroneous – a positive correlation was observed for soil temperature and a negative correlation was observed for soil moisture. Line 283-285 in the results section needs to be modified accordingly.

(ii) Line 308: The citation “Conrad and Seiler, 1985” is not in the reference section.

(iii) L333-335: The authors propose that more frequent soil wetting events in the greywater treatments can lead to shifts in the soil microbiome, nutrient availability, and carbon inputs. Because this statement is supported by unpublished data (V Buzzard, 2021, unpublished data), I expect the authors are well-positioned to propose potential mechanisms explaining benefits of grey water on H2 soil uptake activity before the monsoon precipitations. Are the authors already aware of microbial successions or enrichment of HA-HOB triggered by irrigation treatments?

Reviewer 2 ·

Basic reporting

In this study, Buzzard et al. present observations of H2 fluxes in both managed (Green infrastructure) and unmanaged soil in a semi-arid region.
Both the experimental design and the presentation of the results are sound. This study is suitable for publications provided the following comments are addressed.





line 67
It is estimated that a quarter of H2 emissions are from human activity, disproportionately impacting atmospheric H2 concentrations and the H2 cycle of urban ecosystems.

How did the authors come to their estimate?
This seems a bit low to me. Ehhalt mentions 50% of H2 production is anthropogenic. If we add fossil fuel, anthropogenic activities would contribute >40% to H2 sources.

line 69
I do not understand why understanding the h2 uptake in urban landscape will significantly alter our understanding of the H2 budget. how much land does urban landscape cover?


line 169
could the authors comment on the impact of these different surface layers on H2 uptake.

line 243 - could you indicate which assumptions are not met.

line 250.
I suggest adding a table summarizing the meteorological conditions during the three sampling periods as well as over the last 10 days preceding the sampling. This table should also include information about the soil moisture and soil temperature.

It would also be useful to show the timeseries for soil moisture and soil temperature for the different treatments in Fig. 1.

line 263 and elsewhere

I recommend rephrasing the presentation of the raw statistical output of ANOVA in the text.
For instance, line 263 could be rewritten as

We find that the effect of the green infrastructure (GI) treatment and the
increased wetting events observed during the monsoon season are significant at the 0.05 level and large.

The ges threshold for large/medium/small effects should be indicated in the method section and in Table 1 (similar to the 0.05 threshold used by the authors for the p-value).

line 264 please also report the partial ges values.

line 265 This seems to convey the same results as the previous sentence (same statistics). Please clarify or remove.

line 277

please indicate moisture level for each manipulation.

line 312

the authors imply that the diffusion limited regime for H2 uptake occurs only during soil flooding. This seems at odds with previous studies which indicate that the diffusion limited regime is quite prevalent. Please clarify.

line 322

the decrease of H2 soil flux with temperature is at odds with the reanalysis of previous observations presented in Ehhalt et al. (2011) DOI: 10.1111/j.1600-0889.2011.00581.x which showed a clear increase of HOB activity with temperature for temperature <30C.
It is my understanding that following these results, recent modeling studies have assumed that HA-HOB activity increases with T.

Thus it is important that the authors clarify why their results are different from previously derived T sensitivity of HA-HOB. In addition they need to discuss the potential implications of their results for our understanding of the distribution of H2 uptake and its response to future environmental changes. In particular, it seems that the increase in h2 uptake with climate change that is discussed in line 344 based on Paulot et al. (2021) is not supported by the authors' results.

Minor

line 56

Suggest rephrasing to clarify the connection with GI. Here is a suggestion

The more continuous H2 uptake associated with GI may help reduce the impacts of drought on H2 cycling in semiarid urban ecosystems


line 247

affect -> effect of

line 313

"consistent with research showing that H2 uptake is driven by low moisture levels that activate soil microbes"

I would suggest rephrasing to clarify that under very low soil moisture HOB activity is suppressed.


Fig. 1:
please indicate averaging time period

Fig. 2:
Please use the same acronym in the text (ges) and int the caption (eta_g^2) to refer to the generalized eta square.

Fig. 4 indicate sign convention for H2 flux (atmosphere->soil >0) in caption

Experimental design

No issue

Validity of the findings

No comment

Additional comments

No comment

---

## Round 0.2 · Major Revisions

The authors need to address issues raised by referee.

Reviewer 2 ·

Basic reporting

The authors have addressed most of my comments.
However, as mentioned in my comments to the first draft, the sensitivity to temperature reported in this study differs from the published literature. The authors need to address this difference in the revised manuscript before I can recommend publication.

Major comments:

1) line 52, line 370 and line 492 ; Fig. 4

the R2 between H2 uptake and temperature reported in the abstract has changed from 0.244 to 0.105. Was this a typo in the original manuscript?

I would suggest reporting the correlation rather than R2. The sign of the correlation is important and seems to be different from previous studies. This also applies to line 370.

As mentioned in my previous comments, the sign of the correlation with temperature is at odds with previous studies, which show an increase in soil H2 uptake over the T range considered here (see Fig. 1 of Ehhalt (2013)). It is important that the authors discuss this important difference.
Ehhalt (2013) provides a functional form for the dependence of H2 uptake to temperature. Could the authors add this to Fig. 4a?

Such comparison would also be useful for soil moisture (see Fig. 5 of Ehhalt (2013) or the functional form presented by Yashiro (2011) or Bertagni (2021))


3) Line 392. I suggest that the authors revise the text to better convey that H2 diffusivity into soil is reduced by soil moisture. The text reads as if moisture only impedes H2 uptake under flooded question, which is at odds with the published literature (Yashiro, 2011 ; Ehhalt 2013 ; Bertagni 2021)

Minor comments

1) line 39

"Unlike mesic ecosystems, H2 uptake in arid ecosystems rarely experience diffusion limitation due to water saturated soils and has been shown to increase with soil wetting in arid ecosystems"

This sentence is confusing. Suggests rephrasing

2) Line 109

Could the authors provide the criteria used to define arid and semi-arid regions?

Experimental design

no comment

Validity of the findings

no comment

Additional comments

no comment

---

## Round 0.3 · Minor Revisions

Please revise the manuscript based on the comment from Reviewer 2

Reviewer 2 ·

Basic reporting

I am happy to recommend publication.
See two minor comments below
line 37. remove "limited"
line 49. Please define notation used for correlation, i.e. I believe 62 refers to the number of measurements.

Experimental design

n/a

Validity of the findings

n/a

Additional comments

n/a

---

## Round 0.4 · accepted · Accept

Thank you for responding positively to the comments of the reviewers and for your patience in going through the review process. We appreciate your support of the journal and hope you will publish future work with PeerJ.